# Hybrid Solid-Phase Extraction for Selective Determination of Methamphetamine and Amphetamine in Dyed Hair by Using Gas Chromatography–Mass Spectrometry

**DOI:** 10.3390/molecules24132501

**Published:** 2019-07-09

**Authors:** Nam Hee Kwon, Yu Rim Lee, Hee Seung Kim, Jae Chul Cheong, Jin Young Kim

**Affiliations:** Forensic Genetics & Chemistry Division, Supreme Prosecutors’ Office, Seoul 06590, Korea

**Keywords:** hair analysis, methamphetamine, amphetamine, hybridSPE, GC–MS

## Abstract

Sample preparation is an important step in the isolation of target compounds from complex matrices to perform their reliable and accurate analysis. Hair samples are commonly pulverized or processed as fine cut, depending on preference, before extraction by techniques such as solid-phase extraction (SPE), liquid–liquid extraction, and other methods. In this study, a method based on hybrid solid-phase extraction (hybridSPE) and gas chromatography–mass spectrometry (GC–MS) was developed and validated for the determination of methamphetamine (MA) and amphetamine (AP) in hair. The hair samples were mechanically pulverized after washing with de-ionized water and acetone. The samples were then sonicated in methanol at 50 °C for 1 h and centrifuged at 50,000× *g* for 3 min. The supernatants were transferred onto the hybridSPE cartridge and extracted using 1 mL of 0.05 M methanolic hydrogen chloride. The combined solutions were evaporated to dryness, derivatized using pentafluoropropionic anhydride, and analyzed by GC–MS. Excellent linearity (R^2^ > 0.9998) was achieved in the ranges of 0.05–5.0 ng/mg for AP and 0.1–10.0 ng/mg for MA. The recovery was 83.4–96.8%. The intra- and inter-day accuracies were −9.4% to 5.5% and −5.1% to 3.1%, while the intra- and inter-day precisions were within 8.3% and 6.7%, respectively. The limits of detections were 0.016 ng/mg for AP and 0.031 ng/mg for MA. The validated hybridSPE method was applied to dyed hair for MA and AP extraction and compared to a methanol extraction method currently being used in our laboratory. The results showed that an additional hybridSPE step improved the recovery by 5.7% for low-concentration quality control (QC) samples and by 24.1% for high-concentration QC samples. Additionally, the hybridSPE method was compared to polymeric reversed-phase SPE methods, and the absolute recoveries for hybridSPE were 50% and 20% greater for AP (1.5 ng/mg) and MA (3.0 ng/mg), respectively. In short, the hybridSPE technique was shown to minimize the matrix effects, improving GC–MS analysis of hair. Based on the results, the proposed method proved to be effective for the selective determination of MA and AP in hair samples.

## 1. Introduction

The use of methamphetamine (MA) in illicit drug market has rapidly grown over the years. In East and South-East Asia, MA seizures have increased every year since 2012, measuring up to 60 tons in 2016 according to a United Nations Office on Drugs and Crime (UNODC) report. They were smuggled into the Republic of Korea, Japan, and Australia from Taiwan or Southeast Asian countries [1,2]. The most commonly used illicit drug in the Republic of Korea is MA, locally referred to as ‘philopon’, which accounted for about 52% of drug-related crimes in 2017 [3]. Also, the number of MA users in 2016 increased more than 53% since 2012 [4].

Biological sample matrices that could be analyzed to detect illicit drug use include urine, hair, blood, and oral fluid. These sample matrices provide valuable information regarding drug exposure. Above all, hair analysis has several advantages over urine testing. Generally, hair analysis reveals a relatively long window of drug exposure depending on the length of hair shaft, while urine analysis shows the recent drug exposure. Additionally, drugs can be stored in hair longer than other biological samples, and hair can provide information regarding repeated drug use [5,6,7,8,9,10]. However, hair analysis requires more complex analytical procedures compared to urine analysis. These include decontamination, pulverization of hair, extraction, and instrumental analysis steps [10].

The Society of Hair Testing has guidelines, and recent publications for drug testing in hair indicated that hair treatments must be considered when interpreting the results of a hair drug test [11,12,13]. As the extracted matrix components often interfere with the detection of target compounds, the conditions for extraction and purification of the target compounds from hair need to be optimized to avoid an excessive influence of the interfering substances and further sample clean-up [14,15]. A more careful approach is required for the effective removal of co-extracted compounds from treated hairs such as perm, dyeing, and cosmetic products. Furthermore, significant consideration should be given to the effective elimination of endogenous interferers from the hair matrix if hair pulverization is performed using a mechanical pulverizer [16,17,18].

Because of the high complexity of hair matrix or the presence of contaminant compounds from hair cosmetics, selecting the appropriate sample preparation technique is of greatest importance to make an effective sample clean-up that allows increased detection sensitivity. Sample preparation allows isolating the target compounds from the interfering components in the main matrix and concentrating them for their accurate identification and quantification.

The conventional sample preparation techniques that generally employed are liquid–liquid extraction (LLE) and solid-phase extraction (SPE) [19]. The most common and widespread sample preparation technique is LLE, which is considered the simplest technique for the isolation of the target compound between an aqueous solution and an immiscible organic solvent. Its main drawback is the requirement of large amounts of high-purity solvents that are expensive and toxic and of special handling of organic waste solvents for disposal. Because of the necessity to analyze complex matrices requiring more selectivity than that offered by LLE, SPE emerged as an alternative extraction technique. SPE is becoming more popular than LLE for target compound pre-concentration and matrix removal, due to the large choice of SPE sorbents with highly selective extraction ability, normally producing pure samples [20,21,22]. In this respect, new sorbents based on increased selectivity, such as molecularly imprinted polymers, have been used for SPE to remove matrix interferences in the determination of several drugs of abuse [23,24].

Several methods for the extraction and isolation of amphetamines from hair, removing endogenous or exogenous compounds, have been applied, including LLE [25], SPE [26,27], micropulverized extraction [28], ultrasonic-assisted methanol extraction [29], and headspace solid-phase microextraction [30]. Recently, the hybrid solid-phase extraction (hybridSPE) precipitation technique was introduced to prevent interferences from endogenous proteins and phospholipids present in biological matrices, using a one-step elution method, without affecting the selectivity of SPE. In addition, hybridSPE reduces the levels of interfering components during sample clean-up [21,31].

In the present study, a method based on hybridSPE and gas chromatography–mass spectrometry (GC–MS) was developed and validated for the determination of MA and amphetamine (AP) in hair. Furthermore, the extraction performance of the hybridSPE method was compared to those of other SPE methods. The developed method was also applied to evaluate the performance of hybridSPE for cosmetically treated forensic hair samples.

## 2. Results and Discussion

### 2.1. Method Development

To evaluate the efficiencies of SPE cartridges in the removal of interferents from hair and in the pre-concentration of target compounds, cosmetically treated hair samples obtained from eight MA abusers were pooled, homogenized, and used as a positive control. Figure 1 shows chromatograms for pooled cosmetically treated authentic samples using direct methanol extraction followed by hybridSPE clean up (Figure 1A) or direct methanol extraction only (Figure 1B).

Three different types of SPE cartridges were tested in this study, i.e., two polymeric reversed-phase type SPE cartridges (Phenomenex Strata-X and Waters Oasis HLB) and a Zirconia-based SPE cartridge (HybridSPE, Sigma–Aldrich/Supelco). The relative recovery was calculated from the integrated peak areas using the quantifier ion of each compound (Figure 2). Compared to polymeric reversed-phase SPE methods, the relative recoveries of AP (1.5 ng/mg) and MA (3.0 ng/mg) were 50% and 20% greater, respectively, when using hybridSPE.

In order to choose the optimal concentration of hydrogen chloride (HCl) in methanol as an elution solvent, the concentration of acid was examined within the range of 0.01–0.25 M. Figure 3 shows the optimum concentration of HCl producing the lowest matrix interference in methanol was 0.05 M. The target compounds were influenced by the baseline peaks near the retention time of each compound at higher concentration of HCl. The hybridSPE consisted of zirconia coated on a silica surface and acted as a Lewis acid, so that under acidic conditions, protons effectively protonated carboxylate and phosphate groups in potential interferents and disrupted the binding of negatively charged phosphate or carboxylate groups to zirconia coated on the silica surface, resulting in variations in the removal of interferents from the second fraction [31].

### 2.2. GC–MS Analysis

Chemical derivatization generally leads to improved chromatographic selectivity and peak shape by altering polarity and volatility. Mass spectra for pentafluoropropionic anhydride (PFPA) derivatives of target compounds and internal standard (IS) were generated and characterized by GC–MS. To obtain sufficient sensitivity for the quantitative analysis, the characteristic fragment ions for each compound were selected and monitored. These fragments were determined in a full-scan analysis. Table 1 lists the ions monitored as well as the retention times after PFPA-derivatization of the target compounds. The base peak ion and some of key fragment ions were used to indicate the presence of each compound and IS. These ions were used as quantifier and qualifier ions for the target compounds.

### 2.3. Method Validation

Five different hair samples obtained from non-drug users were examined in a selectivity test. There were no interfering peaks at the retention times of the target compounds and IS (Figure 3). Representative chromatograms obtained from blank hair and spiked hair samples are shown in Figure 4.

The linearity was tested in the concentration ranges of 0.05–5.0 ng/mg for AP and 0.1–10.0 ng/mg for MA. Table 2 presents the calibration parameters such as slope, intercept, and coefficient of determination obtained from the calibration curves (*n* = 4). The R^2^ values for the linear regression were above 0.9998 for each compound, indicating excellent goodness of fit. The calculated limit of detection (LOD) values were 0.016 ng/mg and 0.031 ng/mg, while the lower limit of quantification (LLOQ) values were 0.05 ng/mg and 0.1 ng/mg for AP and MA, respectively.

To evaluate the absence of carry-over, the highest extracted calibrator was injected into the GC–MS instrument, followed by an ethyl acetate blank. Potential carry-over effects were not observed. For practical purpose, ethyl acetate blanks were used throughout the sample sequence to verify that no sample-to-sample contamination occurred.

The recoveries, accuracies, and precisions are summarized in Table 3. The analytical recoveries were determined at three concentration levels in five replicates. The mean recoveries (%) for the target compounds ranged from 87.1% to 96.8% for AP and from 83.4% to 94.8% for MA. The intra- and inter-day precisions were within 8.3% and 6.7%, while the intra- and inter-day accuracies ranged from −9.4% to 5.5% and from −5.1% to 3.1%, respectively. These data were within the acceptance criteria of 15% of nominal concentration for low, middle, and high quality control (QC) concentrations.

Stability experiments were performed to identify any possible variation due to storage time. Low- and high-concentration QC samples were analyzed using a method developed in this study. The results from the stability experiments indicate that the samples were stable under normal storage conditions. No significant loss of the target compounds was observed at room temperature for 20 days.

### 2.4. Forensic Applications

In order to prove the applicability on forensic hair samples, the developed method was applied to analyze cosmetically treated authentic samples obtained from MA users (*n* = 7). Figure 5 presents relative peak areas for MA as well as AP measured using a direct methanol extraction and a direct methanol extraction followed by hybridSPE for cosmetically treated MA-positive hair samples. It clearly indicates that an additional hybridSPE step improved the recovery of AP more than that of MA and provided good isolation and recovery of MA and AP from human hair for subsequent analysis.

The method was used to analyze hair samples obtained from MA users (*n* = 15). Figure 4D shows a typical chromatogram of a cosmetically treated authentic hair sample. In authentic hair samples, AP concentration was in a range of 0.08–4.14 ng/mg with a mean value of 1.06 ng/mg, while MA concentration was in a range of 0.29–9.07 ng/mg with a mean value of 4.12 ng/mg. This method is thus effective in isolating and identifying MA and AP in cosmetically treated hair samples.

## 3. Conclusions

A GC–MS method for the detection and quantification of MA and AP was studied and evaluated in human hair. The use of hybridSPE as a sample preparation technique enabled the effective removal of co-extracted chemical interferences from cosmetic-treated hair and successful extraction and purification of the target compounds from pulverized hair, avoiding an excessive influence of interfering endogenous or co-extracted compounds. The experimental results proved that the proposed method was effective for the selective determination of MA and AP in forensic hair samples. The method based on hybridSPE improved significantly the recovery of MA and AP by, on average, 21.6% and 151.5%, respectively, in cosmetically treated authentic hair samples, compared to a methanol extraction method currently used in our laboratory. The developed method was fully set up for routine detection and quantification of MA and AP in cosmetic-treated hair samples and appears to be a purification technique more effective than those currently used. It would be useful for the removal of endogenous interfering substances or co-extracted compounds from the hair matrix, which will surely provide selective extraction of target compounds from hair prior to instrumental analysis.

## 4. Materials and Methods

### 4.1. Chemicals and Reagents

The reference compounds AP and MA were purchased from Cerilliant (Round Rock, TX, USA) at a concentration of 1000 µg/mL in methanol. Methanolic solutions of the deuterated IS AP-d_8_ and MA-d_11_ in each vial at a concentration of 100 µg/mL were also purchased from Cerilliant. HPLC-grade acetone, ethyl acetate, and methanol were supplied by J. T. Baker (Phillipsburg, NJ, USA). Methanolic HCl (HCl, 1.25 M) was obtained from Fluka (St. Gallen, Switzerland). PFPA was acquired from Acros Organics (Geel, Belgium). HybridSPE phospholipid cartridge (30 mg/1 mL) was purchased from Supelco (Bellefonte, PA, USA). Strata-X 33 µm polymeric reversed phase cartridge (60 mg/3 mL) was purchased from Phenomenex (Torrance, CA, USA), while Oasis HLB extraction cartridge (60 mg/3 mL) was from Waters (Milford, MA, USA). All other chemicals were of analytical grade or higher. Polypropylene tubes (2 mL) were obtained from Eppendorf (Safe-Lock tube, Hamburg, Germany). Water was purified with a Millipore AFS-16 water purification system (Molsheim, France).

Stock solutions of the target compounds were mixed and diluted in methanol to prepare final mixed standard solutions at 10 µg/mL of AP and 20 µg/mL of MA. A working solution of IS (AP-d_8_ and MA-d_11_) at 0.3 µg/mL was also prepared in methanol. All these solutions were stored at −20 °C before use.

### 4.2. Hair Specimens

Blank hair to be used as a matrix for control and calibration samples was obtained from five volunteers. Matrix-matched calibrators were prepared by adding the mixed standard solution to blank hair samples over the concentration ranges of 0.05, 0.1, 0.25, 0.5, 1, 2.5, and 5 ng/mg (AP) and 0.1, 0.2, 0.5, 1, 2, 5, and 10 ng/mg (MA). QC samples were also prepared in the same way, using a separately arranged stock solution at a concentration of 0.15 (low), 1.5 (middle), and 3 ng/mg (high) for AP and 0.3 (low), 3 (middle), and 6 ng/mg (high) for MA.

Authentic hair samples that were collected from MA abusers were acquired from the Narcotics Departments at the District Prosecutors’ Offices in the metropolitan area. Hair samples were usually pulled out or cut as close as possible to the skin from the posterior vertex. Total length was measured, and cosmetic treatments such as coloring and bleaching were noted. All samples were stored under dry, dark conditions at room temperature up to 20 days before analysis.

### 4.3. Sample Preparation

Hair samples were washed with water and acetone according to previously published methods [6]. Then, 10 mg of hair sample was put into a 2 mL polypropylene tube containing six metal beads. Pulverization was performed for 10 min at a frequency of 30 (1/s) (Qiagen TissueLyser II, Retsch, Haan, Germany). The IS solution (50 µL, 0.3 µg/mL of deuterated AP and MA) and 1.0 mL of methanol were added. Each tube was vortexed for 10 s and placed into a 50 °C water bath for 1 h under ultrasonication. The tube was then centrifuged at 50,000× *g* for 3 min to obtain a clear supernatant. In vacuum conditions (Visiprep^TM^ SPE 24 vacuum manifold, Supelco, Bellefonte, PA, USA), 0.8 mL of clear supernatant was passed through a hybridSPE cartridge and collected; then, 1 mL of 0.05 M methanolic HCl was used to sequentially elute the hybridSPE cartridge. These two fractions were combined and concentrated to dryness under a nitrogen stream at 45 °C and 30 kPa using a Caliper TurboVap LV evaporator (Biotage, Harris, NC, USA). The sample was reacted with 50 µL acetone and 50 µL PFPA at 50 °C for 30 min, followed by drying under a nitrogen stream. The residue was reconstituted with 50 µL of ethyl acetate, and a 1 µL aliquot was injected into the GC–MS for analysis.

### 4.4. GC–MS analysis

GC–MS was performed with an Agilent Technologies (Foster City, CA, USA) 5977A mass spectrometer equipped with a 7890B GC and 7693 Autosampler. Chromatographic separation of AP and MA was achieved on a DB-5MS UI capillary column (30 m × 0.25 mm i.d., 0.25 μm film thickness, J&W Scientific, Folsom, CA, USA) with helium as the carrier gas at a ramped flow mode. The flow was 1.1 mL/min for 4.5 min, was increased to 1.25 mL/min at a rate of 0.3 mL/min, was held for 2.4 min, and then was decreased to 1.1 mL/min at a rate of 0.3 mL/min and held for 8 min. The injector temperature was 260 °C, and the GC interface temperature was 280 °C. The oven temperature was initially 90 °C for 1.5 min, was increased to 175 °C at a rate of 15 °C/min, was held for 1.0 min, and then was increased to 300 °C at a rate of 40 °C/min and held for 4.0 min. Total run time was 15.3 min, including 4.5 min of solvent delay time. Splitless injection mode was used with a purge-on time of 0.35 min at a flow rate of 75 mL/min. The mass spectrometer was operated at 70 eV in the electron impact (EI) mode with selected ion monitoring for quantitative analysis. One ion was used as a quantifier, while two other ions were used as qualifiers for the analysis of the target compounds.

### 4.5. Method validation

The developed method in this study was validated by measuring the selectivity, linearity, LOD, LLOQ, carry-over, precision and accuracy, recovery, and stability according to previous protocols [32,33].

To evaluate selectivity, blank hair samples obtained from five different origins were analyzed to determine endogenous compounds or potential interferences released from the matrix. LOD and LLOQ were measured by evaluating the signal/noise (S/N) ratio of 10 replicates of blank hair for each compound at proper concentrations. LOD was calculated on the basis of the concentration with a S/N > 3, while the concentrations of the target compounds with a S/N > 10 were chosen as LLOQ.

The absence of carry-over was evaluated by injecting at the highest point of the calibration curve, followed by solvent blank, and measuring the peak area at the retention times of the target compounds under the investigation. For routine analysis of hair samples, ethyl acetate blanks were run between each pair of samples.

The linearity of the method was evaluated over the concentration ranges of 0.05–5.0 ng/mg for AP and 0.1–10.0 ng/mg for MA and was expressed by the determination coefficient (R^2^). The calibration curves were obtained by least-squares linear regression.

The intra-day precision and accuracy of the method were established by seven independent determinations of the QC samples (*n* = 7). The inter-day precision and accuracy were determined in four different days for the aforementioned replicates over a four-week period. To determine the precision, the coefficients of variations (% C.V.) were calculated for the replicate measurements. Accuracy (% bias) was expressed as the relative error of the calculated concentrations and was calculated by the degree of agreement between the measured and the nominal concentrations of the QC samples.

For recovery determination, QC samples were prepared at three concentration levels. The recovery was determined by comparing the absolute peak area (A) of each compound for the QC samples prepared in five replicates before extraction with the absolute peak area (B) of the target compound for the samples processed as blank and spiked after extraction at the same concentration level. The recovery was calculated using the following equation: Recovery (%) = A/B × 100.

Sample stability was assessed by repeated analysis of low-concentration QC and high-concentration QC samples (*n* = 5), spiked at the above-mentioned concentrations. To examine the stability, QC hair samples were left for 20 days prior to sample preparation and analysis. These samples were analyzed, and the peak area ratios compared with the ones obtained by the analysis of freshly prepared samples. The samples were considered stable if the concentration of the QC samples resulted within ±15% of that of the freshly prepared QC samples.

## Figures and Tables

**Figure 1 molecules-24-02501-f001:**
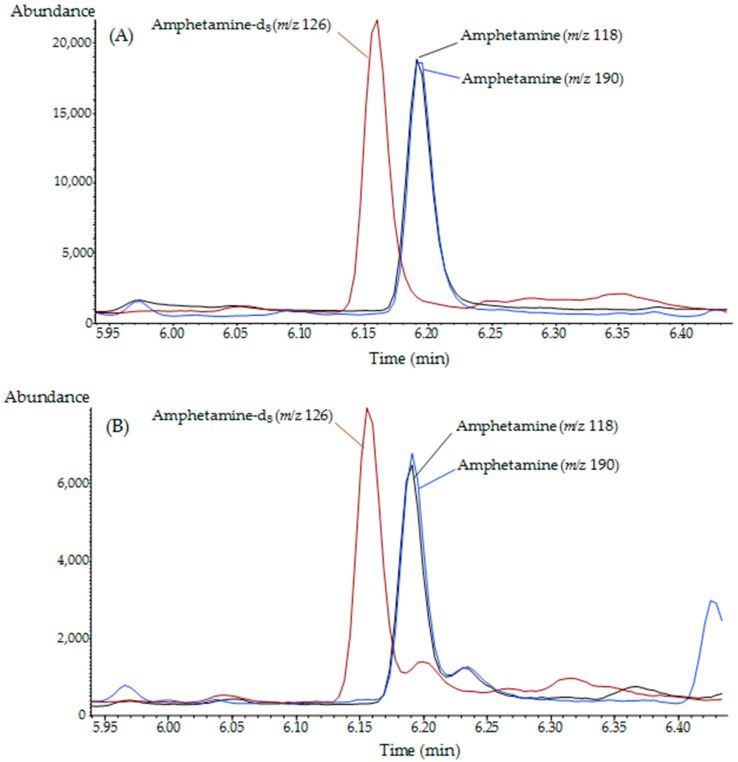
Representative GC–MS chromatograms of cosmetically treated authentic hair samples processed with (**A**) direct methanol extraction–hybrid solid-phase extraction (hybridSPE) and (**B**) direct methanol extraction only.

**Figure 2 molecules-24-02501-f002:**
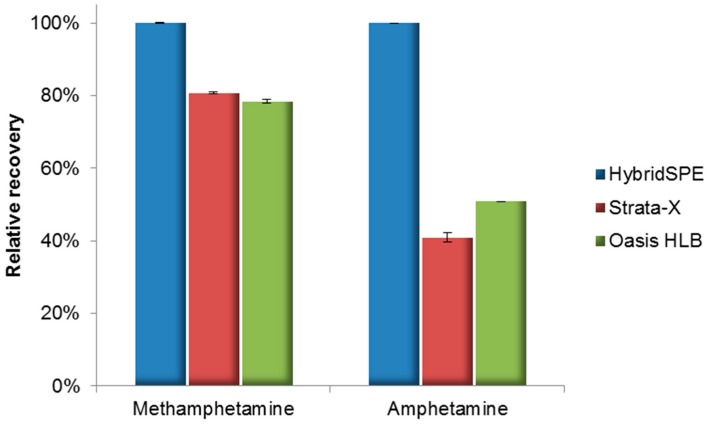
Evaluation of the relative recovery of three different SPE cartridges.

**Figure 3 molecules-24-02501-f003:**
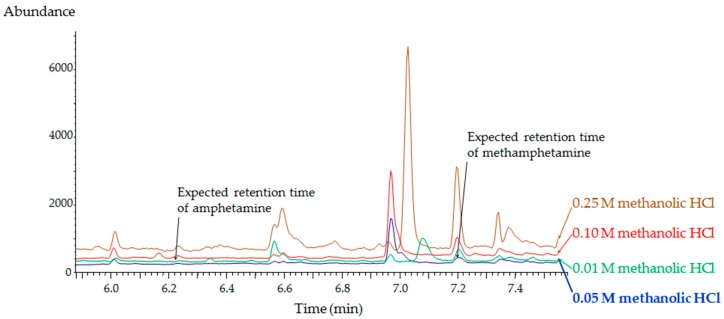
Effects of hydrochloric acid concentration in the eluent on the GC–MS chromatograms of blank hair extracts.

**Figure 4 molecules-24-02501-f004:**
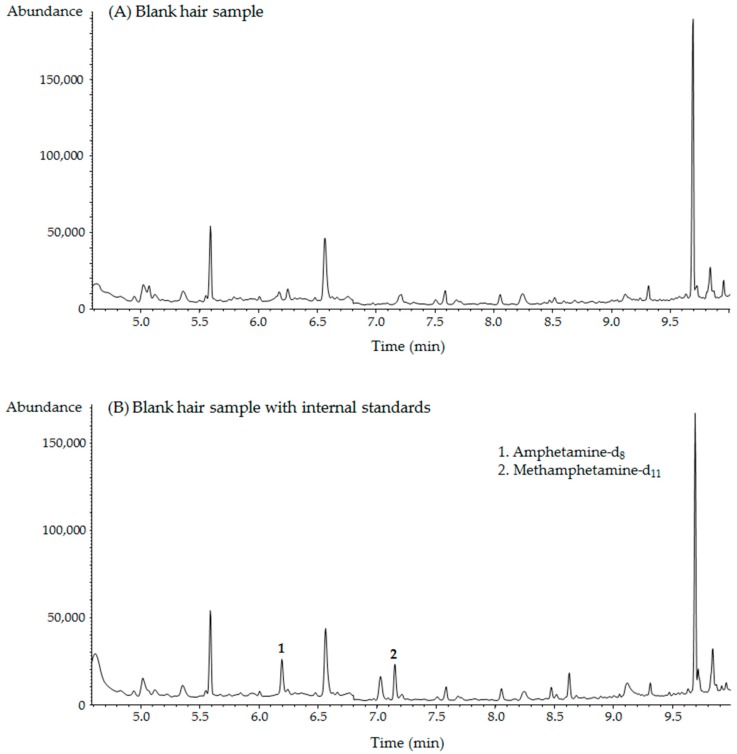
Representative GC–MS chromatograms obtained from (**A**) blank hair, (**B**) blank hair with internal standards, (**C**) spiked hair with 25.0 ng/mg of each compound, and (**D**) cosmetically treated MA-positive hair samples.

**Figure 5 molecules-24-02501-f005:**
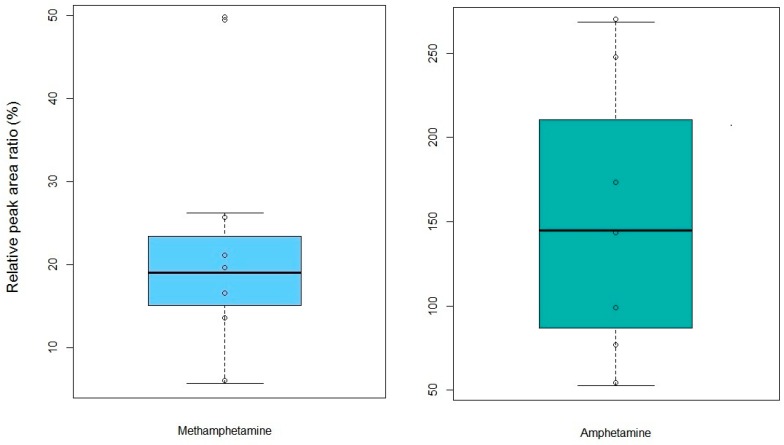
Comparison of relative peak area ratio (%, [A − B]/B) measured using the proposed direct methanol extraction–hybridSPE method (A) and the direct methanol extraction method (B) for the analysis of MA and AP in cosmetically treated MA-positive hair samples.

**Table 1 molecules-24-02501-t001:** Retention times, molecular weights, and ions monitored for GC–MS analysis of pentafluoropropionic anhydride (PFPA) derivatives. AP-d_8_: deuterated amphetamine, MA-d_11_: deuterated methamphetamine.

Compound	Retention Time (min)	Molecular Weight	Ions Monitored (*m/z*)		
			Quantifier Ion	Qualifier Ions	
AP-d_8_	6.19	289	126	-	-
AP	6.22	281	118	190	91
MA-d_11_	7.15	306	210	-	-
MA	7.20	295	118	204	160

**Table 2 molecules-24-02501-t002:** Method calibration.

Compound	Concentration Range (ng/mg)	Slope (Mean ± SD)	y-Intercept (mean)	Linearity ^1^ (R^2^)	LOD ^2^ (ng/mg)	LLOQ ^3^ (ng/mg)
AP	0.05–5.0	1.1968 ± 0.0553	0.0178	0.9998	0.016	0.05
MA	0.1–10.0	0.2826 ± 0.0205	0.0321	0.9999	0.031	0.10

^1^ Linearity is described by the determination coefficient for the calibration curve. ^2^ The limit of detection (LOD) was based on the concentration corresponding to a signal plus 3 standard deviations from the mean of 10 replicates of blank hair. ^3^ The lower limit of quantification (LLOQ) was defined as the lowest concentration on the calibration curve with precision (% CV) less than 20% and accuracy (% bias) within ±20%.

**Table 3 molecules-24-02501-t003:** Results of accuracy, precision, and recovery for method validation using quality control (QC) samples.

Compound	QC (ng/mg)	Recovery (%)	Intra-Day (*n* = 7)		Inter-Day (*n* = 28)	
			Precision ^1^ (% CV)	Accuracy ^2^ (% bias)	Precision (% CV)	Accuracy (% bias)
AP	0.15	87.1	2.2	−9.4	6.7	0.3
	1.5	96.8	1.4	−0.8	3.1	−2.2
	3	91.8	0.4	−4.2	1.8	−5.1
MA	0.3	83.4	8.3	5.5	5.9	3.1
	3	94.8	1.4	−1.0	4.8	−1.0
	6	92.0	0.7	−5.8	3.9	−4.7

^1^ Expressed as the relative standard deviation of the peak area ratios of the target compound/IS. ^2^ Calculated as [(mean calculated concentration − nominal concentration)/nominal concentration] × 100.

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
