# Peer review of "Hybrid Solid-Phase Extraction for Selective Determination of Methamphetamine and Amphetamine in Dyed Hair by Using Gas Chromatography–Mass Spectrometry"

_molecules, 2019, doi:10.3390/molecules24132501_

Round 1

Reviewer 1 Report

The article refers to the simultaneous determination of methamphetamine and amphetamine in dyed hair by GC-MS. Zirconia-based SPE cartridges have been used for sample preparation. The article is well-written and well-organized and the experimental data are clearly presented. To my opinion the paper is useful for the readers and I support its publication after revision.

Comments

1.      Justification of the novelty of the proposed method over the already existing methodology is needed in the introduction section

2.      A critical discussion of the results in relation to the already published reports is needed in the conclusion section

3.      Data presented in Table 4, could be better included in a diagram.  

4.      References should be increased; suggested references:

https://www.ncbi.nlm.nih.gov/pubmed/15291265

https://www.ncbi.nlm.nih.gov/pubmed/19745539

https://www.tandfonline.com/doi/abs/10.1080/00365510802439072?journalCode=iclb20

https://www.ncbi.nlm.nih.gov/pubmed/18288687

https://www.ncbi.nlm.nih.gov/pubmed/18585989

Author Response

Thank you for your helpful advice.

Reviewer 2 Report

The manuscript is interesting considering the importance of employed analytical methods to determine methamphetamine and amphetamine illicit drugs in hair matrices. The authors have optimized different aspects of the extraction method. The advantage is clearly presented in the work. It is understandable, and it is well organized. And the authors present an interesting comparison of extraction procedure of these analytes. I find no problems with the scientific approach or technical content presented by the authors in this manuscript.

Author Response

We thanks for your careful reading of the manuscript, and especially for your helpful comments.

Reviewer 3 Report

The manuscript proposed herein by NH. Kwon et al. deals with the development and validation of an analytical method based on hybrid solid-phase extraction (hybridSPE) and gas chromatography-mass spectrometry (GC-MS) for the determination of methamphetamine (MA) and amphetamine (AP) in hair, after a derivatisation step. Moreover, hybridSPE was compared to polymeric reversed-phase SPE cartridges in terms of performace and the method was applied to cosmetically treated MA-positive forensic hair samples.

The experimental design is presented in a fairly clear manner as well as figures and tables, while validation is complete.

However, unfortunately the present Reviewer finds it difficult to understand the elements of novelty that can justify the publication of this work.

The scientific literature on the subject is rich in effective methods for the determination of MA and AP in hair matrix, either alone or in combination with other amphetamines or even together with other classes of drugs of abuse. The instrumental methods used by other authors exploit both liquid chromatography (HPLC) and gas chromatography (GC), the latter complicated by the fact that it requires a step of derivatization, therefore representing a possible source of variability.

SPE-based pretreatment strategies are among the most widely used for purification of extracts resulting from the incubation of hair matrices. Although the Authors take into consideration a variant recently introduced in scientific research, represented by HybridSPE, this has already been extensively tested for the simultaneous determination of 75 drugs abuse (including several amphetamines) in human hair samples. HybridSPE performances were compared with other pretreatment procedures (namely filtration, dispersive SPE and liquid-liquid extraction) [Shin Y. et al., Biomed Chromatogr., 2019, 22:e4600].

As regards method application, the Authors state that their main objective is to verify the effectiveness of the pretreatment approach when applied to cosmetically treated positive hair samples. Although a comparison was done between HybridSPE and untreated methanolic extract, and between HybridSPE and other types of SPE cartridges, no comparison is provided between cosmetically treated positive samples and non-cosmetically treated positive samples. Therefore, it is not possible to assess the actual superiority of the proposed method with respect to what is already present in the literature. Furthermore, no consideration is made about the effects that such cosmetic treatments can have on the concentrations of the analytes in the matrix. Such assessments have already been widely performed in already published works [Agius R., Drug Test Anal., 2014, 1:110-119].

Considering these aspects, the present Reviewer believes that the work proposed herein does not contain sufficient elements of novelty to justify its publication on Molecules. The same instrumental approach is already applied for the analysis of the same compounds (and several other analogues) in the same matrix, the pretreatment procedure proposed by the Authors has already been considered in a similar work and the method application not completely relevant to the purpose the Authors aim at. For these reasons, this manuscript should be rejected.

Author Response

We thanks for your comments.

Reviewer 4 Report

The authors present an improved method based on hybrid solid phase extraction for the determination of  methamphetamine and 3 amphetamine in dyed hair by using gas chromatography–mass spectrometry. The manuscript is well written and contains a good level of detail regarding the analytical method and results. I feel that the paper fits well with the scope of the journal so could be published if the authors can address the following points

Abstract - a good level of detail overall, but the beginning of the abstract needs to better introduce the main purpose of the paper. The current sentence "Efficient sample preparation technique is essential for the selective detection of analytes." is not a good introductory statement

The introduction is quite short and does not contain that many references, I think there could be a better description of current methods of extraction and detection for these compounds and the limitations of the current methods, so this better justifies the new approach. There is some information presented by the authors but it has to stand out more clearly.

Results and Discussion

Figure 3 is a bit small and this doesn't aid interpretation

Figure 4. needs to be better annotated and full names of abbreviations given in legend to aid interpretation

There needs to be a summary of the discussion which compares the advances in the current method to the previous published methods. If there are any future validation studies required or further development this should also be outlined i.e. what is required to make this method useful in forensic analysis, or is it already developed sufficiently

Conclusion needs a bit more detail in line with the abstract

materials and methods this is well written and structured with a good level of detail

Author Response

Thank you for your helpful advice.

Round 2

Reviewer 3 Report

Considering the changes made and the sections implemented in the manuscript, together the Authors' rebuttals to the notes raised by this Reviewer, the manuscript proposed herein by NH. Kwonet al. can now be considered adequate for publication in Molecules.